# Appliance-Level Anomaly Detection by Using Control Charts and Artificial Neural Networks with Power Profiles

**DOI:** 10.3390/s22176639

**Published:** 2022-09-02

**Authors:** Hanife Apaydin-Özkan

**Affiliations:** Department of Electrical and Electronics Engineering, Eskisehir Technical University, 26555 Eskisehir, Turkey; hapaydin1@eskisehir.edu.tr

**Keywords:** IoT, home appliance, power profile, Artificial Neural Network, Control Chart, anomaly detection

## Abstract

Nowadays, the development of the Internet of Things (IoT) concept has increased the interest in some technologies, one of which is the detection of anomalies in home appliances before they occur. In this work, in order to contribute to the works that use appliance power profiles for anomaly detection, a novel Appliance Monitoring and Anomaly Detection System (AM-ADS) is presented. AM-ADS consists of a main controller, a database, IoT-based communication units, home appliances, and power measurement units (smart plugs or special measurement equipments) mounted on appliances. In AM-ADS, a new Control Chart (CC) based method, for the cases that a limited number of historical power profiles are available; and a new Artificial Neural Network (ANN) based method, for the cases that a sufficient number of historical power profiles of each anomaly free and anomalous situations are available, are used according to the developed rule-based AM-ADS procedure to maximize the advantages and to eliminate the disadvantages of these methods as much as possible. According to the AM-ADS procedure, power consumptions of appliances, which provide meaningful information about the health of appliances, are measured during their operations and the corresponding power profiles are created. Active power, power factor, and operation duration features of power profiles are considered as decisive control parameters and different characteristics of these parameters are used as inputs for CC and ANN-based methods. The efficiency and performance of AM-ADS are validated by application case studies, where the ability to detect anomalies varies between 94.56% and 99.03% when a limited number of historical data is available; and the ability to detect and classify anomalies varies between 96.34% and 99.45% when a sufficient number of historical data is available.

## 1. Introduction

In recent years, parallel to the development of internet and sensor technologies, the concept of the Internet of Things (IoT) has entered our lives. Within the IoT concept, many appliances, that we use in our daily lives, have become communicable with each other and with other internet-connected devices, as well as some appliances acquire information about their environment and usage situations. Moreover, appliances gain features that can update their status information by communicating with people. Hence, people gain opportunities to observe and control them from anywhere, anytime. These opportunities have brought many technologies to increase energy management, safety, and user comfort, that make human life easier. One of these technologies still being studied is the detection of anomalies in appliances before they occur.

Because misuse and aging effects in appliances may cause some anomalies that increase power consumption and decrease durability as well as create serious safety problems such as fire and electric shock that will endanger human life. Early detection of anomalies in appliances can help to avoid these undesirable situations and to reduce after-sale expenses for both users and manufacturers, and also provides improvements in energy efficiency. Therefore, anomaly detection in appliances has been given increasing attention.

Many studies propose to use power profiles of appliances for detecting anomalies. In some of these studies, power consumption is monitored at the aggregated level by a common meter located on the home’s main supply. But, it is difficult to associate an anomaly with an appliance as common power monitoring is performed. Therefore, appliance level power monitoring, thus the meters are located at the plug level and the power consumption of each appliance is monitored individually, is more widely used for anomaly detection of appliances.

Appliance-level power monitoring based works in the literature are composed of supervised and unsupervised anomaly detection methods. Unsupervised methods detect anomalies by labeling power profiles as anomaly free or anomalous according to certain rules and methods without the need for much historical data [1]; while supervised methods use machine learning classifiers for detection and classification of appliance anomalies when there exist sufficient numbers of historical anomaly free and anomalous power profile data.

In this work, in order to contribute to the works in the direction of appliance level power monitoring for anomaly detection and classification, a novel Appliance Monitoring and Anomaly Detection System (AM-ADS) using both a supervised method and an unsupervised method according to a rule-based procedure is presented.

In this work, for detecting anomalies, an unsupervised method based on the Control Chart (CC) which is the most effective tool of Statistical Process Control (SPC) to monitor and detect unusual fluctuations that arise in the outputs, is developed to be used in AM-ADS for the cases that a limited number of historical data is available.

In the developed CC-based anomaly detection method, upper and lower limits, namely control limits, and the corresponding control intervals of decisive features of power profiles are determined from the anomaly-free historical power profiles of appliances. CCs of decisive properties of each appliance are constructed according to its control intervals. Anomaly detection of an appliance is performed by placing and checking decisive features of the present power profile at the corresponding CC. In the CC-based method, only appliances’ anomaly-free power profiles are stored in a database and only these data are used to determine the control limits of decisive features. Hence, this method can be used even when few historical data is available. To the best of the author’s knowledge, a CC-based method is used for anomaly detection of appliances for the first time in the literature.

In this work, for both detecting and classifying anomalies, a supervised method based on Artificial Neural Networks (ANNs) is developed to be used in AM-ADS for the cases that a sufficient number of historical data is available. The relationship between the power profiles of appliances and their anomalies, which AM-ADS uses basically, has a complex and non-linear structure. ANNs can learn and model the relationships in that structure. Besides, ANNs do not impose any constraints on input variables (like how they should be distributed) as different from the other classification methods. For these scientific reasons, in AM-ADS, the use of ANN is preferred for anomaly detection and classification.

In the developed ANN-based anomaly detection and classification method, the ANN model of each appliance is specifically designed. However, every configuration is feedforward and with a single hidden layer trained by backpropagation. Input layers of constructed ANNs consist of features extracted from power profiles, while the output layers present the anomaly-free situation and the anomalous situations defined for the belonging appliance. Hidden layers are designed experimentally due to the training performance of ANNs. The developed ANN-based method provides a much more diverse and comprehensive anomaly classification compared to similar works in the literature as it can detect not only user usage anomalies but also component anomalies.

The proposed AM-ADS in this work consists of a main controller, a database, IoT-based communication units, home appliances, and power measurement units (smart plugs or special measurement equipments) mounted on appliances. As described above, AM-ADS in this work uses both CC and ANN-based methods according to the developed rule-based AM-ADS procedure to maximize the advantages and eliminate the disadvantages of these methods as much as possible.

According to the AM-ADS procedure, power consumptions of appliances are measured during their operations and sent to the main controller in predefined small time intervals. The main controller creates corresponding power profiles and stores them in a database as historical power profiles. In AM-ADS, active power, power factor, and operation duration features are extracted from the power profiles of appliances and considered as decisive control parameters for anomaly detection. ANN-based method of AM-ADS uses maximum, minimum, and mean values of these parameters as inputs. CC-based method of AM-ADS defines upper-lower control limits and the corresponding control intervals of these parameters to construct the control chart.

At the end of the AM-ADS procedure, if no anomaly is detected at the current operation, the power profile is labeled as anomaly free; otherwise, it is labeled as anomalous with the type of anomaly (if it is known). The labeled power profile is stored in the database, and the relevant decisive parameters are updated according to the newly added data. The efficiency and performance of AM-ADS are validated by application case studies.

The main contributions of AM-ADS to the literature can be summed up as follows:AM-ADS is a realistic method since it is based on real power profiles of appliances.AM-ADS is usable independent of the number of historical power profiles.To the best knowledge of the author, a CC-based method is used for anomaly detection of appliances for the first time in the literature.Developed ANN-based method provides a much more diverse and comprehensive anomaly classification compared to similar works in the literature as it can detect not only user usage anomalies but also component anomalies.AM-ADS provides very high accurate results; such that, the CC-based method (used when a limited number of historical data is available) detects anomalies between 94.56% and 99.03% accuracy; while the ANN-based method (used when sufficient numbers of historical power profiles of each anomaly free and anomalous situations are available) detects and classifies anomaly between 96.34% and 99.45% accuracy.Every appliance regardless of its brand and type can be included in AM-ADS simply with only a smart plug connection without making any mechanical changes in the appliance itself.

## 2. Literature Survey

Studies on anomaly detection in appliances have gained acceleration in recent years. Many studies use power profiles of appliances for detecting anomalies. These power profiles are obtained by monitoring power consumptions at the aggregated level or at the appliance level.

In the works using aggregated level monitoring, anomaly detection is performed by various methods such as presenting a hierarchical probabilistic model [2], an online learning-based intelligent algorithm [3], a classification algorithm [4], a fuzzy rule-based intelligent identification method [5], a fast event detection algorithm [6] and a K-means clustering algorithm [7]. Furthermore, in [8] a method is proposed to save energy by detecting AC faults in advance, while abnormal energy use of the refrigerator and air conditioner is tracked for anomaly detection in [9]. However, it is difficult to associate an anomaly with an appliance by using aggregated level monitoring. Therefore, the appliance-level power monitoring approach is more preferred for anomaly detection of appliances. Note that, this method is used in many different applications besides anomaly detection: in [10], a near-real-time plug load identification method is developed for several office plug loads, while another device identification method based on the information obtained from the plug-meter is introduced in [11]. Appliance level power monitoring is also used for detecting occupancy, occupant movement and user-appliance interaction [12,13].

In the literature, appliance-level power monitoring for anomaly detection consists of supervised and unsupervised procedures.

Unsupervised procedures use several methods, such as k-means clustering [14], fuzzy clustering [15] and one class-neural network [16] etc. Furthermore, in [17], a graphical visualization tool is proposed for supporting the detection of power consumption anomalies using a rule-based approach, while a waveform feature extraction model for anomaly detection in power profiles is proposed in [18]. For the same purpose, ref. [19] proposes a rule-based model comparing actual energy consumption and the predicted one. In these studies, it is possible to detect the anomaly in the appliance without the need for much historical data, but the anomaly can not be classified. Note that, in this work, a CC-based method is proposed as an unsupervised procedure. Although this method is used for anomaly detection of appliances for the first time in the literature, it has been widely used in industrial applications and production processes. For example, in [20], authors use the CC method to control the production parameters of machine production, while this method is used for service request management of a help desk in [21] and for controlling the resistance-layer thickness of integrated circuits in [22]. In [23,24], different dimensions of various mechanical parts are evaluated and the production parameters of the machine that produces these parts are tried to be kept under control via CCs.

On the other hand, numerous works in the literature present supervised procedures using machine learning classifiers for anomaly detection and classification of appliances. The deep convolutional neural network component introduced in [25] identifies the nonperiodicity of electricity theft and the periodicity of normal electricity usage. In [26], anomaly detection is performed by using linear and non-linear regression methods by using artificial neural networks and an autoregressive integrated moving average model. The study [27] proposes deep learning algorithms that have the capability of removing seasonality and trend from the power profile data, while [28] proposes a model to identify and localize the detected anomalies by using a combination of neural network and K-means algorithms. In another study [29], a hybrid model using recurrent neural networks and quantitative regression is introduced to predict and detect abnormal power consumption. These methods detect anomalies as well as classify them with high accuracy, but a sufficient number of anomalous and anomaly-free power consumption data are needed for these methods.

As different from the literature, in AM-ADS, both unsupervised and supervised procedures based on CC and ANN methods are used for anomaly detection and/or classification. The advantages of AM-ADS compared to the studies in the literature are explained in the Introduction section.

## 3. Appliances

A typical living environment contains many home appliances. These appliances consist of active (motor, fan, etc.) and passive (heater, etc.) components. Each appliance may have several program modes offering different functionality and features, such as operation durations, energy consumptions, and more. They perform their functions by activating one or more of these components [30,31]. As a result, the power consumption of an appliance at any given moment is the aggregated consumption of all these currently activated components.

In AM-ADS, the set of appliances is represented by L and the set of program modes of an appliance *a* is represented by $M^{a}=\left\{1,2,...\right|M^{a}\left|\right\}$ ($|M^{a}|\geq 1$), while the operation duration of a program mode $j\in M^{a}$ of an appliance *a* is indicated by $n^{a_{j}}$ which is discretized into prescribed uniform internal time slots, i.e, $\hat{t}\in D^{a_{j}}=\left\{1,\;2,\;,...,\;d^{a_{j}}\right\}$. Here, $d^{a_{j}}=n^{a_{j}}\Delta_{t}$ where $\Delta_{t}$ represents the length of each internal time slot $\hat{t}$. A *power profile* of an appliance *a* for a program mode *j*, i.e., $P^{a_{j}}\in R^{d^{a_{j}}\times 1}$, is obtained as the view of its power consumption in the course of the operation. The set of appliances L can be divided into two subsets: variable power appliances (thus, $P^{a_{j}}\left(\hat{t}\right)$ is time-varying) the set of which is represented by $L_{v}$ and fixed power appliances (thus, $P^{a_{j}}\left(\hat{t}\right)$ has negligible variations in time) the set of which is represented by $L_{f}$). Note that, $L=L_{f}\cap L_{v}$.

Note that, the power profiles of appliances with cooling and/or heating functions (washing machine, dishwasher, refrigerator, air conditioner and etc.) may be affected by environmental conditions (ambient, water temperature, etc.). For the rest, power profiles of a program mode of an appliance are similar at every operation. Any considerable difference in power profiles is considered as an early sign of an anomaly within an electrical component of the considered appliance or anomalous user usage.

Some of the most used household appliances and their most seen anomalies considered in this work are described below.

### 3.1. Refrigerator and Air Conditioner

Refrigerator (Ref) and Air Conditioner (AC) are multi-functional and variable power appliances with several electrical components, i.e., Ref, AC $\in L_{v}$. Basic electrical components of Ref are the compressor, freezer fan, fresh food fan, and defrost heater, while those of AC are the compressor, outer unit fan, inside unit fan, and defrost heater. Power profiles of these appliances are dominated by the compressor since its power consumption is much more than that of other components. The compressor switches between on-off states periodically during the operation of the appliance. Hence, these compressor-containing appliances have periodic power profiles.

The most common component anomalies in Ref and AC are compressor anomalies and fan anomalies which affect power profiles of these appliances. As an example, power profiles of a Ref for anomaly-free and fresh food fan anomalous situations are given in Figure 1. As it is clear from the figure, the power consumption increases and becomes unstable in the anomaly situation.

On the other hand, Ref and AC may be subject to anomalous user usage. The most common user usage anomalies of AC are open room window and blocking filter anomalies, while those of Ref are open-door for a long time and overloading anomalies which affect the power profiles of these appliances. For example, high power consumption duration increases in the open-door anomaly situation since the compressor runs longer to cool inside when the door is open.

### 3.2. Washing Machine and Dishwasher

Washing Machine (WM) and DishWasher (DW) are also multi-functional and variable power appliances with several electrical components, i.e., WM, DW $\in L_{v}$. Basic electrical components of WM are the heater, drum motor, and drain pump; while those of DW are the heater, circulation pump, and drain pump. Each of these components has significantly different power consumptions. Besides, these appliances have different program modes (such as long, express, regular and etc.) due to the laundry or dish to be washed. Therefore, power profiles of these appliances depend on activated program modes [30].

The most common component anomalies in WM and DW are heating element, drain pump, rotating part (for WM), and circulation motor (for DW) anomalies. The power profile of the long program mode of WM for anomaly-free and rotating part anomaly situations are given in Figure 2. As it is clear from the figure, the power consumption during the motor rotation (while the drain shaft is operating) increases due to the increase in mechanical losses (See Figure 3 for enlarged view).

On the other hand, WM and DW may be subject to user usage anomalies. The most common user usage anomalies of WM and DW are overloading and underloading. When WM is overloaded, the power consumption increases since the washer adds extra stress to the motor and the tub bearings.

### 3.3. Iron, Kettle, and Lamp

Iron, Kettle, and Lamp are single-functional (e.g., kettle, lighting, etc.) and fixed power appliances with a single active electrical component, i.e., Iron, Kettle, Lamp $\in L_{f}$. In these appliances, belonging electrical components can have anomaly or long time operation can lead to anomalous situation. Power profiles of a Lamp for the anomaly-free situation and anomalous situation (some LEDs are broken) are given in Figure 4. As it is clear from the figure, power consumption of the Lamp decreases in the anomalous situation due to broken LEDs.

## 4. Technical Background

In this study, two different appliance anomaly detection methods based on Control Chart and Artificial Neural Networks are used for anomaly detection and/or classification. The background of these techniques will be explained in this section.

### 4.1. Statistical Process Control-Control Chart

Statistical Process Control (SPC) which is widely used in industrial applications and production processes, is a technique that uses statistical tools to analyze a process or its outputs to control, manage, and improve the quality of the output or the capability of the process [21]. Control Chart (CC) (or Shewhart chart) is the most effective tool of SPC to monitor the performance of the process and to detect unusual fluctuations that arise in the process [32].

In CC, measurements are taken regarding the parameter to be monitored and controlled in the process and the corresponding CC of these measurement data is created.

The most common statistically determined control limits for a data set $X=\{x_{1},\;x_{2},\;...,x_{n}\}$ used in CC are lower limit $lc_{{lim}_{x}}$ and upper limit $uc_{{lim}_{x}}$ which are calculated as follows [22]:(1)$$\begin{array}{c} lc_{{lim}_{X}}=\mu_{X}-3\sigma_{X}\\ uc_{{lim}_{X}}=\mu_{X}+3\sigma_{X} \end{array}$$

Here $\mu_{X}$ is the mean and $\sigma_{X}$ is the standard deviation of *X*. The corresponding control interval of the data set *X* is determined as follows:(2)$$\Delta^{c}X=\left[lc_{{lim}_{X}}\;uc_{{lim}_{X}}\right]$$

If one member of a data set *X* is out of the determined control interval it is labeled as anomalous. According to the statistical knowledge [33], the probability of the output of an ordinary process falling outside this control interval is 0.27%.

As an example, in Figure 5, the output (colored blue) of a process, with its CC is represented. One output point of the process is out of the control interval, thus anomalous. Hence the health of the process must be examined.

### 4.2. Artificial Neural Network

Artificial Neural Network (ANN) which is a soft-computing tool that can learn patterns and predict, is applied to many engineering problems effectively because of its capability to provide fast, reliable, and accurate solutions to nonlinear problems easily.

ANN is organized in layers; such as an input layer, a number of hidden layer(s), and an output layer. These layers are made up of a number of interconnected nodes, namely neurons. The number of hidden layers and neurons in the layers may vary according to the problem studied. A typical feed-forward multilayered neural network, which consists of three layers such as an input layer (with *n* input nodes), one hidden layer (with *m* hidden nodes), and an output layer (with *k* output nodes), is given in Figure 6.

ANNs are trained by using training data sets until a predefined terminating condition is met for the error. Then the accuracy and performance of the trained ANN can be estimated by testing the data set.

BackPropagation Algorithm (BPA) is the most commonly used training algorithm for feed-forward multilayered ANNs. It is based on propagating the input in the forward direction and backpropagating the output in the backward direction by updating the weights correspondingly until a predefined terminating condition is met for the error. Detailed information about BPA can be found in [35,36].

## 5. Appliance Monitoring and Anomaly Detection System

In this study, an Appliance Monitoring and Anomaly Detection System (AM-ADS), which provides early detection and classification of anomalies in household appliances by utilizing both a new CC-based method and a new ANN-based method according to a rule-based procedure, is proposed. AM-ADS consists of the Main Controller (MC), a database, IoT-based communication units, home appliances, and power measurement units (smart plugs or special measurement equipments) mounted on appliances.

AM-ADS is based on the fact that for an appliance any anomaly in any component or anomalous user usage affects the power consumption characteristics of the considered appliance. Within this context, active power, power factor, and operation duration features of power profiles of appliances are considered as decisive control parameters in AM-ADS. CC-based method of AM-ADS uses upper-lower control limits and the corresponding control intervals of these parameters for detecting anomalies, while the ANN-based method of AM-ADS uses maximum, minimum, and mean values of these parameters for detecting and classifying anomalies.

AM-ADS procedure, whose flowchart is given in Figure 7, consists of data acquisition, working mode detection, power profile creation, program mode detection, data classification, and anomaly detection steps respectively. The general procedure applied to an appliance *a* is explained below.

**Data acquisition**: Power consumption of the appliance *a* is monitored during its operation and the smart plug connected to the appliance measures its active power ($p^{a}\left(\hat{t}\right)$), voltage ($v^{a}\left(\hat{t}\right)$), current ($i^{a}\left(\hat{t}\right)$) and power factor ($pf^{a}\left(\hat{t}\right)$), and sends these data to MC at each internal time slot $\hat{t}$.

**Working Mode Detection**: MC detects the working mode, i.e., *on* (operating), *standby* and *off*, of an appliance *a* due to the following rule:(3)$$WMode\left(\hat{t}\right)=\left\{\begin{array}{ccc} On & , & if\;p^{a}\left(\hat{t}\right)\geq p_{std}^{a},\\ Standby & , & if\;p^{a}\left(\hat{t}\right)>0\wedge p^{a}\left(\hat{t}\right)<p_{std}^{a},\\ Off & , & if\;\;p^{a}\left(\hat{t}\right)=0, \end{array}\right.$$

Here, $p_{std}^{a}$ is the maximum stand-by power of appliance *a*.

**Power profile creation**: If *a* is operating, MC creates the power profile $P^{a}\in R^{d_{a}\times 1}$ and power factor vector $PF^{a}\in R^{d_{a}\times 1}$ of the appliance *a* for its operation duration $d^{a}$, as follows
(4)$$P^{a}\left(\hat{t}\right)=p^{a}\left(\hat{t}\right)\;\;\;\;\;\hat{t}\in \left\{0,...,d^{a}\right\}$$
(5)$$PF^{a}\left(\hat{t}\right)=pf^{a}\left(\hat{t}\right)\;\;\;\;\;\hat{t}\in \left\{0,...,d^{a}\right\}$$

Note that, operation durations of some appliances (i.e., TV, lamp, kettle, iron and etc.) depend on user preferences, while that of some (i.e., WM, DW and etc.) depends on the running program mode. For example, active power evaluation of a Ref during a day is given in Figure 8.

**Program Mode Detection**: MC determines the program mode $m^{a}\in M^{a}$ of the appliance by comparing the minimum and maximum values of the duration of its program modes with the present operation duration (length of the power profile).
(6)$$m^{a}=\left\{\begin{array}{ccc} 1 & , & if\;d^{a}<=\frac{d_{max}^{a_{1}}+d_{min}^{a_{2}}}{2},\\ 2 & , & if\;d^{a}>\frac{d_{max}^{a_{1}}+d_{min}^{a_{2}}}{2}\wedge d^{a}<\frac{d_{max}^{a_{2}}+d_{min}^{a_{3}}}{2};\\ .. & .. & ..\\ n & , & if\;\;d^{a}>=\frac{d_{max}^{a_{n}-1}+d_{min}^{a_{n}}}{2}, \end{array}\right.$$

Here, $d_{max}^{a_{j}}$ and $d_{min}^{a_{j}}$ are the maximum and minimum operation durations of a program mode $j\in M^{a}$, respectively.

**Data classification**: In AM-ADS, in order to make a precise analysis, members of power profiles of appliances are divided into parts, namely power modes, according to their values. The set of power modes of an appliance *a* is represented by $Pm^{a}$.

Variable power appliances $a\in L_{v}$ have three power modes such as low,med,high, i.e., $Pm^{a}=\left\{low,med,high\right\}$, while fixed power appliances $a\in L_{f}$ has only fix power mode, i.e., $Pm^{a}=\left\{fix\right\}$

For power mode $i\in Pm^{a}$ and program mode $j\in M^{a}$ of a variable power appliance $a\in L_{v}$, the power profile set, i.e., $P_{i}^{a_{j}}$, the power factor set, i.e., $PF_{i}^{a_{j}}$, and the duration parameter, i.e., $d_{i}^{a_{j}}$, are constructed as follows: $$\begin{array}{c} P_{low}^{a_{j}}=\left\{P^{a_{j}}\left(i\right)|\min_{i}\left\{P^{a_{j}}\right\}<P^{a_{j}}\left(i\right)<\frac{\mu_{P^{a_{j}}}+\min_{i}\left\{P^{a_{j}}\right\}}{2}\right\}\\ PF_{low}^{a_{j}}=\left\{PF^{a_{j}}\left(i\right)|P^{a_{j}}\left(i\right)\in P_{low}^{a_{j}}\;,\;i\in \left\{1,2,..,\right|P^{a_{j}}\left|\right\}\right\} \end{array}$$
with the size of $d_{low}^{a_{j}}=\;|P_{low}^{a_{j}}|\;=|PF_{low}^{a_{j}}|$.
$$\begin{array}{c} P_{med}^{a_{j}}=\left\{P^{a_{j}}\left(i\right)|\frac{\mu_{P^{a_{j}}}+\min_{i}\left\{P^{a_{j}}\right\}}{2}<P^{a_{j}}\left(i\right)<\frac{\mu_{P^{a_{j}}}+\max_{i}\left\{P^{a_{j}}\right\}}{2}\right\}\\ PF_{med}^{a_{j}}=\left\{PF^{a_{j}}\left(i\right)|P^{a_{j}}\left(i\right)\in P_{med}^{a_{j}}\;,\;i\in \left\{1,2,..,\right|P^{a_{j}}\left|\right\}\right\} \end{array}$$
with the size of $d_{med}^{a_{j}}=\;|P_{med}^{a_{j}}|\;=|PF_{med}^{a_{j}}|$.
$$\begin{array}{c} P_{high}^{a_{j}}=\left\{P^{a_{j}}\left(i\right)|\frac{\mu_{P^{a_{j}}}+\max_{i}\left\{P^{a_{j}}\right\}}{2}<P^{a_{j}}\left(i\right)<\max_{i}\left\{P^{a_{j}}\right\}\right\}\\ PF_{high}^{a_{j}}=\left\{PF^{a_{j}}\left(i\right)|P^{a_{j}}\left(i\right)\in P_{high}^{a_{j}}\;,\;i\in \left\{1,2,..,\right|P^{a_{j}}\left|\right\}\right\} \end{array}$$
with the size of $d_{high}^{a_{j}}=\;|P_{high}^{a_{j}}|\;=|PF_{high}^{a_{j}}|$.

For a fixed power appliance $a\in L_{f}$, corresponding sets of program mode $j\in M^{a}$ are as follows:$$\begin{array}{ccc} P_{fix}^{a_{j}}=P_{fix}^{a_{j}}=P^{a_{j}} & PF_{fix}^{a_{j}}=PF^{a_{j}} & d_{fix}^{a_{j}}=d^{a_{j}} \end{array}$$

For example, classified power profile of a Ref, whose power profile is given in Figure 8, is represented in Figure 9.

**Anomaly detection**: AM-ADS consists of two methods for detection and/or classification of anomalies: CC-based method that can detect anomalies when a limited number of historical power profiles is available and ANN-based method that can detect and classify anomalies when sufficient numbers of historical power profiles of each anomaly free and anomalous situations are available.


**Control Chart method for Anomaly Detection (CC-AD):**
Anomaly control parameters considered in CC-AD are active power, power factor, and operation duration of power profiles. In this work, control intervals of these parameters are calculated according to Equation (Equation 2) for each power mode by using the historical anomaly-free power profiles.For program mode $j\in M_{a}$ and power mode $i\in Pm^{a}$ of an appliance a∈L, control intervals of decisive parameters are defined as follows:$\Delta^{c}P_{i}^{a_{j}}:=\left[{lc_{lim}}_{P_{i}^{a_{j}}}\;{uc_{lim}}_{P_{i}^{a_{j}}}\right]$ is the power control interval where ${lc_{lim}}_{P_{i}^{a_{j}}}$/${uc_{lim}}_{P_{i}^{a_{j}}}$ represents lower/upper power control limit$\Delta^{c}PF_{i}^{a_{j}}:=\left[{lc_{lim}}_{PF_{i}^{a_{j}}}\;{uc_{lim}}_{PF_{i}^{a_{j}}}\right]$ is the power factor control interval where ${lc_{lim}}_{PF_{i}^{a_{j}}}$/ ${uc_{lim}}_{PF_{i}^{a_{j}}}$ represents lower/upper power factor control limit,$\Delta^{c}D_{i}^{a_{j}}:=\left[{lc_{lim}}_{d_{i}^{a_{j}}}\;{uc_{lim}}_{d_{i}^{a_{j}}}\right]$ is duration control interval where ${lc_{lim}}_{d_{i}^{a_{j}}}$/${uc_{lim}}_{d_{i}^{a_{j}}}$ represents lower/upper control limit of duration.If an appliance a∈L is anomaly-free, control parameters of its operation outputs satisfy the following three conditions by 99.7%$\forall j\in M^{a}$ and $\forall i\in Pm^{a}$:iEach power value at each power mode of the present power profile is in the corresponding control interval:
(7)$$\begin{array}{cc} P_{i}^{a_{j}}\left(\hat{t}\right)\in \Delta^{c}P_{i}^{a_{j}} & \forall \hat{t}\in D_{i}^{a_{j}} \end{array}$$iiEach power factor value at each power mode of the present power profile is in the corresponding control interval:
(8)$$\begin{array}{cc} PF_{i}^{a_{j}}\left(\hat{t}\right)\in \Delta^{c}PF_{i}^{a_{j}} & \forall \hat{t}\in D_{i}^{a_{j}} \end{array}$$iiiDuration of operation at each power mode of the present power profile is in the corresponding control interval:
(9)$$\begin{array}{c} d_{i}^{a_{j}}\in \Delta^{c}D_{i}^{a_{j}} \end{array}$$For the present operation of an appliance, if any control parameter of the corresponding power profile is out of the corresponding control interval, this means an anomalous situation.For example, CCs for med power mode of Ref, whose classified power profile is given in Figure 9, is given in Figure 10. As seen in the figure, some points of med power mode are out of the control interval of this mode. This indicates that there is an anomaly with one of the active components in this power mode.
**ANN Method for Anomaly Detection and Classification (ANN-ADC):**
In ANN-ADC, feedforward ANNs with a single hidden layer trained by BPA are used for every appliance; however, the ANN of each appliance is specifically designed and has a different configuration. Configured appliance ANNs are trained by historical power profiles of anomaly-free and anomalous situations. Hence, for training, a sufficient number of historical power profiles of each situation must be available in the database to apply ANN-ADC.For each power mode, maximum, minimum, and mean values of active power, as well as power factor and the operation duration are fixed inputs of appliance ANNs. Thus, the input layer of ANN of each appliance a∈L has $|M^{a}|$ number of input nodes for representing the active program mode $\hat{j}\in M^{a}$ and also has $|Pm^{a}|\times 7$ number of input nodes assigned to the 7 characteristics of the output of the operation for every power mode $i\in Pm^{a}$ of the present program mode $\hat{j}$. That is, maximum amplitude in $P_{i}^{a_{\hat{j}}}$ ($P_{i_{max}}^{a_{\hat{j}}}$), mean amplitude of $P_{i}^{a_{\hat{j}}}$ ($P_{i_{mean}}^{a_{\hat{j}}}$), minimum amplitude in $P_{i}^{a_{\hat{j}}}$ ($P_{i_{min}}^{a_{\hat{j}}}$), maximum amplitude in $PF_{i}^{a_{\hat{j}}}$ ($PF_{i_{max}}^{a_{\hat{j}}}$), mean amplitude of $PF_{i}^{a_{\hat{j}}}$, ($PF_{i_{mean}}^{a_{\hat{j}}}$), minimum amplitude in $PF_{i}^{a_{\hat{j}}}$ ($PF_{i_{min}}^{a_{\hat{j}}}$) and the operation duration $d_{i}^{a_{\hat{j}}}$. Hence, every ANN has $|Pm^{a}|\times 7+M^{a}$ number of unchanged input nodes. If the power profile of *a* is affected by the season (i.e., Ref, AC, kettle, WM, DW), its ANN model has 1 more input node, such as $|Pm^{a}|\times 7+M^{a}+1$.The output layer of appliance ANNs presents anomaly types of the corresponding appliance by 1 and 0 which specify whether the corresponding anomaly has occurred or not, respectively. The number of output nodes of an ANN depends on the anomaly types defined for the appliance it belongs to. Hidden layers of appliances are designed experimentally due to the training performance of designed ANNs.For example, the lamp is a fixed power appliance (i.e., it has only one power mode) and its power profile is independent of the season, therefore ANN model of lamp (lamp-ANN) has seven input nodes ($P_{max}^{lamp}$,$P_{min}^{lamp}$,$P_{mean}^{lamp}$,$PF_{max}^{lamp}$,$PF_{min}^{lamp}$,$PF_{mean}^{lamp}$,$d^{lamp}$). On the other hand, the lamp has two anomaly types, such as, led element and long time operation anomalies. Consequently, the lamp-ANN configuration consists of three output nodes representing long-time operation and component anomalous situations and anomaly-free situation. The hidden layer of this ANN is configured with 6 hidden neurons due to the training performance.As another example, the ANN of Ref (Ref-ANN) has 22 input nodes, since Ref has 3 power modes and its power profile is affected by the season, and has 6 output nodes since 5 types of anomaly are considered for Ref. It is configured with 20 hidden nodes due to the training performance. When the extracted features of the power profile given in Figure 1 are applied, the ANN results in a Fresh Food Fan anomaly as expected.In Table 1, anomaly types and the corresponding ANN configurations are given for the most used appliances.**Result evaluation**: If an anomaly is detected by the CC-AD method or detected and classified by the ANN-ADC method, it is reported to the home user and also technical service and/or energy provider (depending on the type of anomaly). If no anomaly is detected at the considered power profile, it is stored in the database as a historical anomaly-free power profile, and the corresponding control intervals are updated accordingly; otherwise, it is stored as an anomalous power profile with its anomaly type if it is known.

The corresponding algorithm of the AM-ADS procedure is given as the AM-ADS algorithm (Algorithm 1) which calls the CC-AD Algorithm (Algorithm 2) if a limited number of historical power profiles are available and ANN-ADC Algorithm (Algorithm 3) when sufficient numbers of historical anomalous and anomaly free power profiles are available.
**Algorithm 1:** AM-ADS Algorithm**Data**: *a*,$P^{a}$,$PF^{a}$,season**Result**: Normal operation or detected anomaly
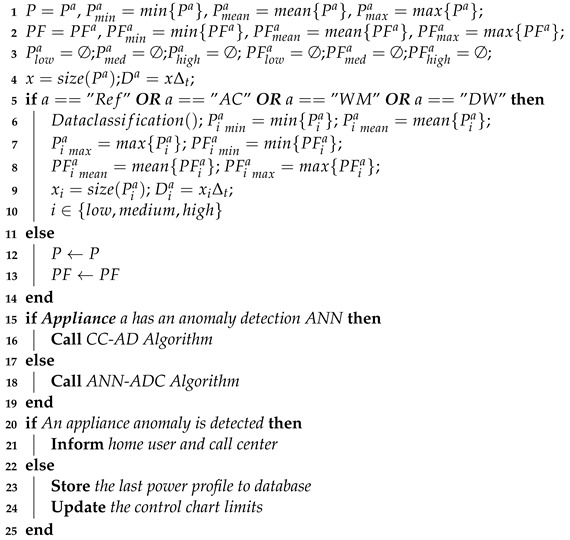


**Algorithm 2:** CC-AD Algorithm
**Input**: *a*, $P^{a}$, $PF^{a}$, $P_{i}^{a}$, $PF_{i}^{a}$, season,$P_{min}$, $P_{mean}$, $P_{max}$, $PF_{min}$, $PF_{mean}$, $PF_{max}$, ${P_{i}^{a}}_{min}$, ${P_{i}^{a}}_{mean}$, ${P_{i}^{a}}_{max}$,${PF_{i}^{a}}_{min}$, ${PF_{i}^{a}}_{mean}$, ${PF_{i}^{a}}_{max}$, $D^{a}$, $D_{i}^{a}$,*x*, $x_{i}$,i∈{low,medium,high};
**Output**: Normal operation or detected anomaly


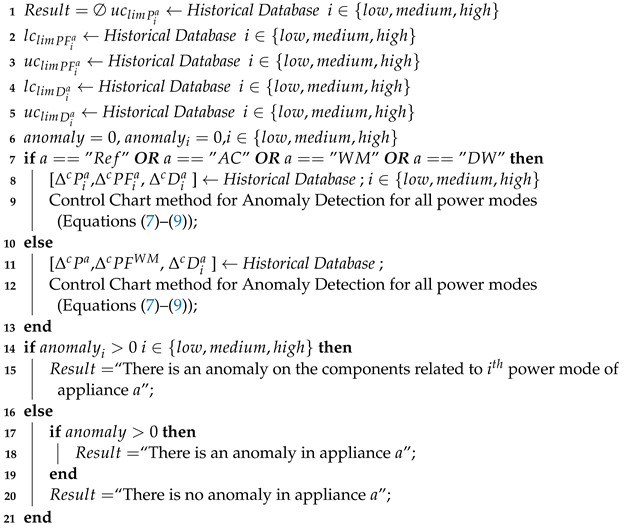




**Algorithm 3:** ANN-ADC Algorithm
**Input**: *a*, season,$P_{min}$, $P_{mean}$, $P_{max}$, $PF_{min}$, $PF_{mean}$, $PF_{max}$, ${P_{i}^{a}}_{min}$, ${P_{i}^{a}}_{mean}$, ${P_{i}^{a}}_{max}$, ${PF_{i}^{a}}_{min}$, ${PF_{i}^{a}}_{mean}$, ${PF_{i}^{a}}_{max}$, $D^{a}$, $D_{i}^{a}$, *x*, $x_{i}$, $PM_{j}^{a}$, i∈{low,medium,high};
**Output**: Normal operation or detected anomaly


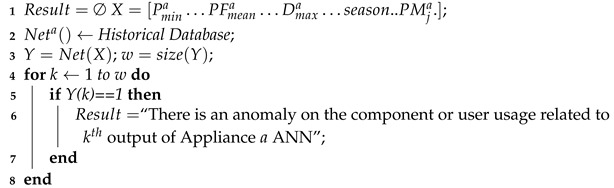




## 6. Case Study and Discussion

For analysing the reliability and performance of the proposed AM-ADS system, several experiments and simulation studies were carried out. In the experiments, the most common home appliances (i.e., Ref, AC, WM, DW, kettle, lamp, and iron) with their most common anomaly types are considered (see Table 1).

In order to generate the power profiles of appliances for experiments, various anomalies were created in the laboratory environment. Power profiles of component-based anomalies were created by intervening in the mechanics or structure of the components, such that distorting the shaft eccentricity, blocking the fan blades, blocking the drain pipe and etc. User usage anomalies were created by realizing anomalous user usage, such that opening the door for a long time, blocking the filter of AC, overloading Ref, WM, DW and etc.

For appliances with multiple program modes (such as WM and DW), anomalous and anomaly-free operations were repeated for each program mode. For appliances that are affected by environmental factors (Ref, AC, WM, DW, and kettle) these operations were carried out repeatedly in both summer and winter seasons as well.

In order to demonstrate the performance of applied methods, an appropriate diagnostic test is applied to the results of experiments. Although the detection of anomalies in the appliances has previously stopped a minor issue from becoming overwhelming, the first goal of an anomaly testing system is to correctly diagnose the anomalous event. Therefore, the accuracy metrics to be used are important in evaluating the results of anomaly detection. The metrics to be used in this study are given below.

accuracy represents the percentage of correctly detecting the values and is calculated as follows:
(10)$$accuracy=\frac{tp+tn}{tp+fp+tn+fn}$$Here, tp describes true positives (number of correct detection of anomalous cycles), fp explains false positives (number of false detection of a normal cycle as anomalous), tn defines true negatives (number of true detection of normal cycles), and fn expresses false negatives (number of false detection of an anomalous cycle as normal).specificity refers to the probability of a negative test.
(11)$$specificity=\frac{tn}{tn+fp}$$F1−score is specific to anomaly detection methods and varies in the range [0 1]. The higher the score, the better the performance of the algorithm. The per-class value of F1-score is
(12)$$F1-score=2\times \frac{precision\times recall}{precision+recall}$$
where $precision=\frac{tp}{tp+fp}$ and $recall=\frac{tp}{tp+fn}$

In the case of multi-class classification, averaging techniques are applied to F1−score calculation, resulting in different averaged scores as macro, weighted and micro averaged F1−scores [37,38,39].

Macro-Averaging: The macro-averaged F1−score is calculated by Equation (Equation 12) while $precision=\frac{\sum_{i=1}^{m}\frac{tp_{i}}{tp_{i}+fp_{i}}}{m}$ and $recall=\frac{\sum_{i=1}^{m}\frac{tp_{i}}{tp_{i}+fn_{i}}}{m}$. Here, *m* is the number of classes; $tp_{i}$, $fp_{i}$ and $fn_{i}$ are the number of true positives, false positives and false negatives of class *i* respectively. So, the macro-averaged F1−score is the arithmetic mean of all the per-class F1−score.Weighted-Averaging: The weighted-averaged F1−score is calculated by Equation (Equation 12) while $precision=\sum_{i=1}^{m}\kappa_{i}\times \frac{tp_{i}}{tp_{i}+fp_{i}}$ and $recall=\sum_{i=1}^{m}\kappa_{i}\times \frac{tp_{i}}{tp_{i}+fn_{i}}$. Here, $\kappa_{i}=\frac{n_{i}}{n}$ is the weighting coefficient, where $n_{i}$ is the number of individual samples labeled by $i_{th}$ class, *n* is the total number of samples. So, the weighted-averaged F1−score takes the mean of all per-class F1−score while considering the number of actual occurrences of the class in the dataset.Micro-Averaging: The micro-averaged F1−score is calculated by Equation (Equation 12) while $precision=\frac{\sum_{i=1}^{m}\left(tp_{i}\right)}{\sum_{i=1}^{m}\left(tp_{i}+fp_{i}\right)}$, $recall=\frac{\sum_{i=1}^{m}\left(tp_{i}\right)}{\sum_{i=1}^{m}\left(tp_{i}+fn_{i}\right)}$. So, the micro-averaged F1−score computes a global average F1−score by counting the sums of the true positives, false negatives, and false positives.

Note that, in the case of an imbalanced dataset where all classes are equally important, it is wise to use macro-averaged F1−score. Hence, in the proposed AM-ADS that treats all classes equally, macro-averaged F1-score is considered for performance evaluation.

Let we consider WM experiments in detail. A 7 kg front-load WM with 3 basic program modes; such as, $M^{WM}=\left\{regular\left(1\right),long\left(2\right),express\left(3\right)\right\}$, are used in these experiments. The power profile of the anomaly-free WM operating at program mode 2 is given in Figure 11a and its CC is given in Figure 11b.

For the CC-AD method, CC parameters of program mode 2 of WM for winter are given in Table 2. In winter, parameters of normally operating WM at program mode 2, $P_{i}^{{WM}_{2}}$, $PF_{i}^{{WM}_{2}}$, $D_{i}^{{WM}_{2}}$ vary within the control intervals $\Delta^{c}P_{i}^{{WM}_{2}}$$\Delta^{c}PF_{i}^{{WM}_{2}}$$\Delta^{c}D_{i}^{{WM}_{2}}$$\forall i\in Pm^{a}$ during its operation.

The power profile of an anomalous WM operating at the long program mode is represented in Figure 12a and its CC is given in Figure 12b. As seen in the figure, some power values measured in the med power mode are out of the control interval which indicates anomalous operation.

The confusion matrix of the CC-AD method obtained by 80 WM experiments is given in Figure 13a. According to this matrix, accuracy is 95.00%, specificity is 1.00, precision is 0.94, and recall is 0.96 and F1−score is 94.79% in WM experiments using CC-AD method. specificity=1 means that WM would not be erroneously classified as anomalous.

When it comes to the ANN-ADC method, the input layer of WM-ANN (Figure 14) has $M^{WM}=3$ number of input nodes for representing the active program mode and has $|Pm^{WM}|\times 7=21$ number of input nodes assigned to the 7 characteristics of the power profile for every power mode $Pm^{WM}$, and 1 input node representing the season (0 or 1 representing winter or summer). The output layer of WM-ANN has 6 nodes representing anomaly-free situations and Heater (Anomaly 1), Rotating part (Anomaly 2), Drain pump (Anomaly 3), Underloading (Anomaly 4), and Overloading (Anomaly 5) anomalies.

For the experiments of the ANN-ADC method, about 75% of the created power profiles in a laboratory environment are used for training while about 25% of them are used for testing.

The confusion matrix of the ANN-ADC method obtained by 240 experiments is given in Figure 13b. By using this confusion matrix, accuracy, precision, recall and F1−score values of each anomaly class are calculated and given in Table 3. The overall accuracy of detecting the anomalies of WM by using the ANN-ADC method is 98.89%. The proposed method can detect component-related anomalies without error. F1−scores of anomaly classes vary in 89.50%–100%. The averaged F1−scores are between 96.67% and 96.93%, while the macro-avg. F1−score, which we took into account, is found to be 96.93%. As seen in the table, Anomaly 4 of WM has the lowest F1−score (89.50%). Because some power profiles with Anomaly 4 are very similar to those of an anomaly-free WM, AM-ADS is more likely to classify them as anomaly free.

In order to evaluate the performance of AM-ADS, CC-AD and ANN-ADC methods are performed both separately and as a hybrid. In the hybrid case, only anomaly detection is performed, such that both ANN-ADC and CC-ADC methods are examined for each power profile, and the results are labeled as anomaly and anomaly-free regardless of the anomaly class determined by the ANN-ADC method.

In Table 4, accuracy and F1−score values of CC-AD, ANN-ADC, and hybrid methods are given for all appliances. As seen from the table, the CC-AD method, which is examined with a limited number (40 to 70) of power profiles, provides F1−scores between 93.45% and 98.79% for anomaly detection of appliances; the ANN-ADC method, which is examined with a numerous number of power profiles (200 to 280) provides F1−scores in between 95.65% and 99.15% for anomaly detection and classification; while the hybrid approach provides F1−scores between 94.22% and 98.92% for anomaly detection when the same power profiles as in ANN-ADC experiments are used. According to these results, as it is expected, anomaly detection performance of the hybrid method is better than the anomaly detection performance of CC-AD and worse than the anomaly classification performance of ANN-ADC.

## 7. Conclusions

Contemporary advancements in IoT technology increased the number of works on many new technologies that facilitate daily human life. One of these technologies still being studied is the early detection of appliance anomalies. In this work, a novel and realistic Appliance Monitoring and Anomaly Detection System, namely AM-ADS, based on appliance level power monitoring for anomaly detection and classification is presented.

In AM-ADS, active power, power factor, and operation duration features are extracted from power profiles of appliances and they are considered as decisive control parameters for anomaly detection and/or classification and analyzed by using two newly developed methods, thus CC-AD method and ANN-ADC method, according to the rule-based AM-ADS procedure.

CC-AD method is used for the cases that a limited number of historical power profile data is available and defines upper-lower control limits and the corresponding control intervals of these parameters to detect an anomaly. To the best of the author’s knowledge, a CC-based method is used for anomaly detection of appliances for the first time in the literature.

ANN-based anomaly detection and classification method is used for the cases that a sufficient number of historical power profiles of each anomaly-free and anomalous situations are available. In this method, the ANN model of each appliance is specifically designed, and the maximum, minimum and mean values of decisive control parameters are defined as the inputs of the ANN models to detect and also classify anomalies. ANN-ADC method provides a much more diverse and comprehensive anomaly classification compared to similar works in the literature as it can detect not only user usage anomalies but also component anomalies.

The major advantage of AM-ADS is detecting anomalies regardless of the number of historical power profile data since both supervised and unsupervised methods are used according to the circumstances. AM-ADS provides very high accurate results; such that, the CC-AD method used when a limited number of historical data is available detects anomaly with 94.56%–99.03% accuracy; while the ANN-ADC method used when a sufficient number of historical power profiles is available detects and classifies anomaly with 96.34%–99.45% accuracy. On the other hand, since AM-ADS uses decisive parameters of power profiles of appliances for detecting and/or classifying anomalies, every appliance, whose power profile can be monitored, can be included in AM-ADS. By foreseeing faults in appliances before they occur, AM-ADS can help to avoid increased power consumption, decreased durability, and serious safety problems. AM-ADS can also act as a useful tool for manufacturers and service organizations for the assessment of the durability of home appliances. Moreover, integrating AM-ADS in manufacturers’ R&D activities can support the potential implementations contributing to the prevention of premature obsolescence of home appliances.

As future directions of this work, different learning methods (i.e., k-means, density-based spatial clustering ) and devices other than home appliances will be integrated into AM-ADS. Furthermore, it is planned to design a rule-based control system that interferes (via interrupting or stopping) with the operation of appliances according to some specific criteria depending on the nature of the detected anomaly and the safety problems it may cause.

## Figures and Tables

**Figure 1 sensors-22-06639-f001:**
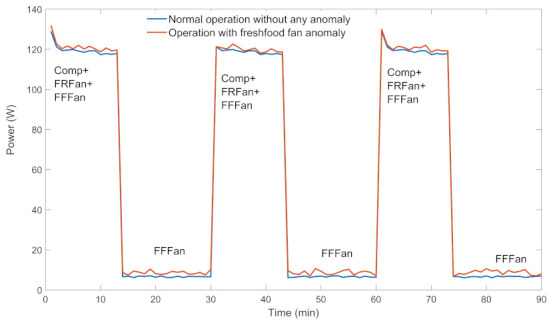
Power profiles of Ref with /without anomaly.

**Figure 2 sensors-22-06639-f002:**
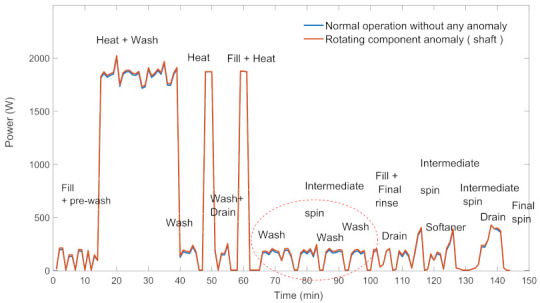
Power profiles of WM with/without anomaly.

**Figure 3 sensors-22-06639-f003:**
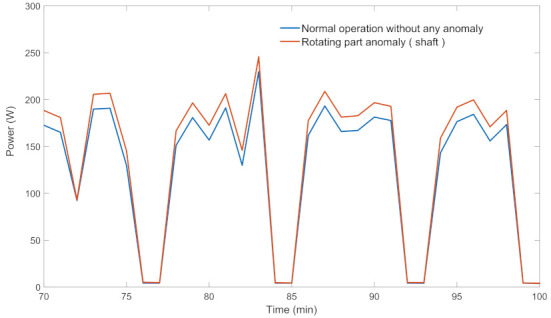
Enlarged view of power profiles of WM with/without anomaly.

**Figure 4 sensors-22-06639-f004:**
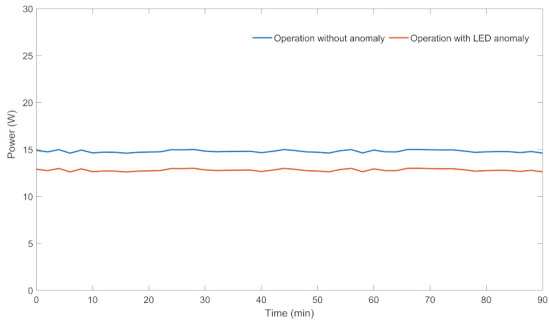
Power profiles of a lamp for anomalous and anomaly-free situations.

**Figure 5 sensors-22-06639-f005:**
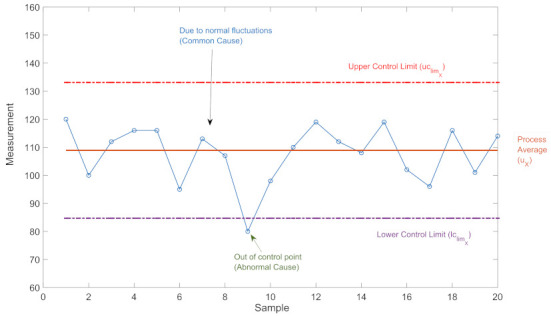
CC of the output of a process [34].

**Figure 6 sensors-22-06639-f006:**
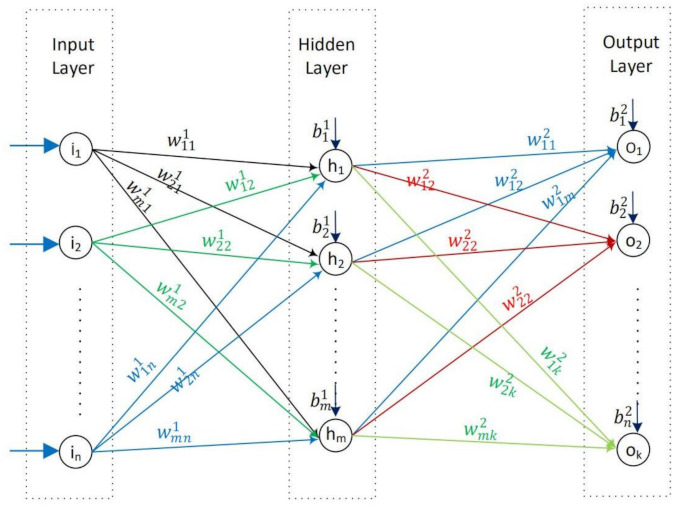
ANN-General.

**Figure 7 sensors-22-06639-f007:**
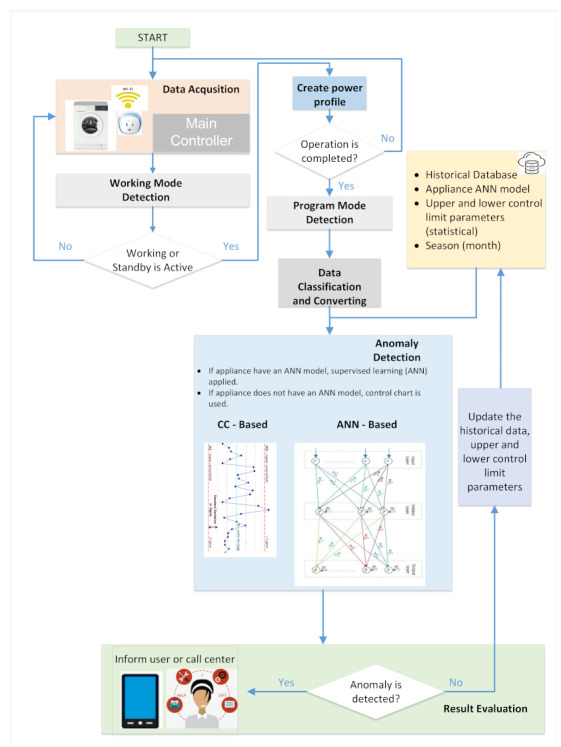
The flowchart of AM-ADS procedure.

**Figure 8 sensors-22-06639-f008:**
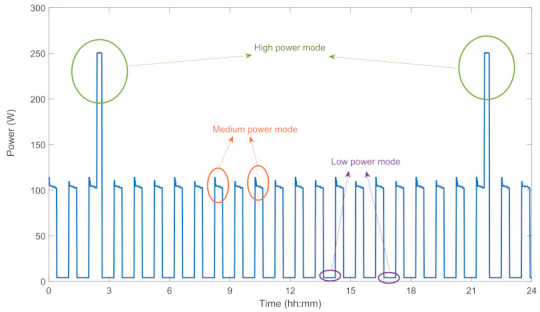
Power profile of a Ref.

**Figure 9 sensors-22-06639-f009:**
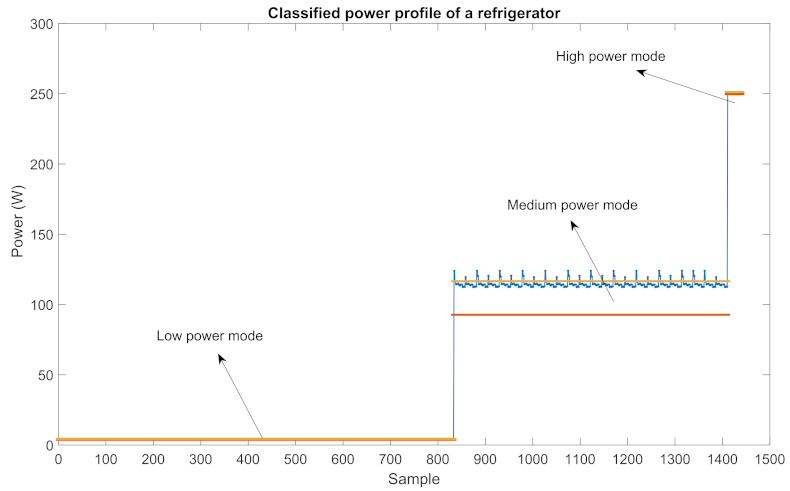
The classified power profile of a Ref.

**Figure 10 sensors-22-06639-f010:**
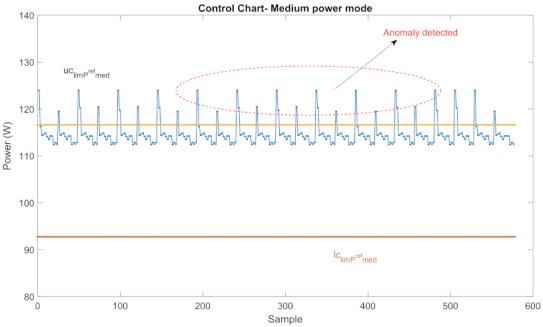
Control limits of a Ref for med power mode.

**Figure 11 sensors-22-06639-f011:**
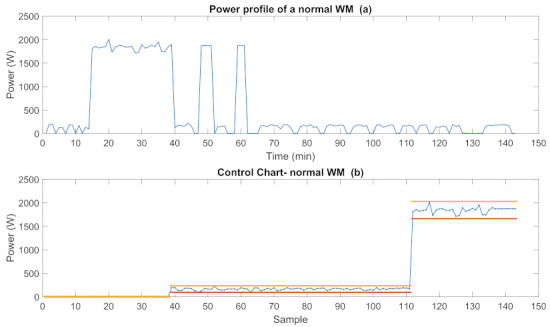
Power profile and CC of an anomaly-free WM.

**Figure 12 sensors-22-06639-f012:**
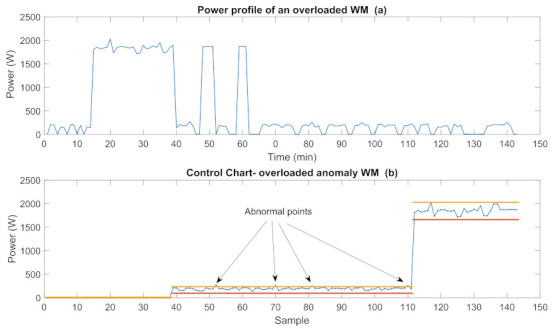
Power profiles and CC of an overloaded WM.

**Figure 13 sensors-22-06639-f013:**
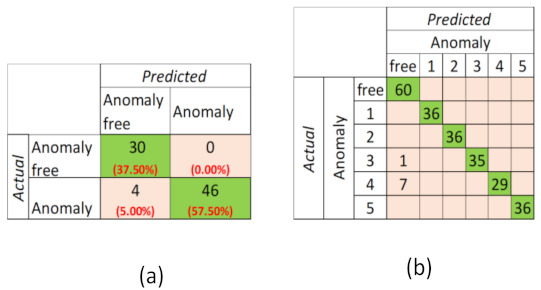
Confusion matrix of WM experiments (**a**) via CC-AD/(**b**) via ANN-ADC.

**Figure 14 sensors-22-06639-f014:**
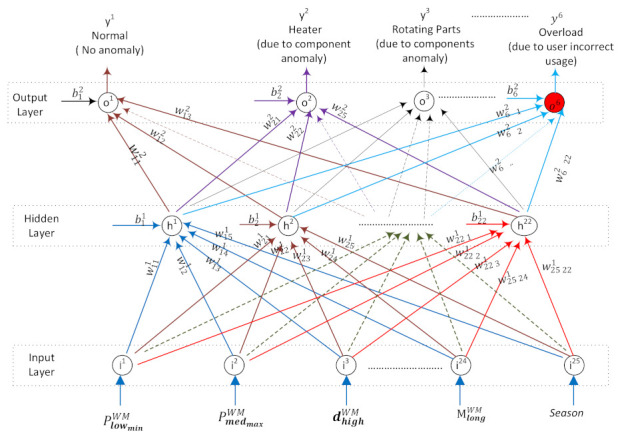
WM-ANN model.

**Table 1 sensors-22-06639-t001:** Anomaly types and the ANN configurations of some appliances.

Appliance	Component Anomalies			User Usage Anomalies		$n_{i}$-$n_{h}$-$n_{o}$
WM	Heater	Rotating parts	Drain pump	Underload	Overload	25-22-6
DW	Heater	Circulation motor	Drain pump	Underload	Overload	25-22-6
Ref	Compressor	FRF	FFF	Open door	Overload	22-20-6
AC	Compressor	Interior fan	Outer fan	Open window/door	Blocking filter	22-20-6
Kettle	Heater/Thermostat			Long time operation		7-6-3
Lamp	Led			Long time operation		7-6-3

**Table 2 sensors-22-06639-t002:** Control intervals of CC-AD method for program mode 2 of WM

Low power mode control intervals for program mode 2	
$\Delta^{c}P_{low}^{{WM}_{2}}=\left[{lc_{lim}}_{P_{low}^{{WM}_{2}}}\;{uc_{lim}}_{P_{low}^{{WM}_{2}}}\right]$	[0.69 10.16]
$\Delta^{c}PF_{low}^{{WM}_{2}}=\left[{lc_{lim}}_{PF_{low}^{{WM}_{2}}}\;{uc_{lim}}_{PF_{low}^{{WM}_{2}}}\right]$	[0.4646 0.5851]
$\Delta^{c}d_{low}^{{WM}_{2}}=\left[{lc_{lim}}_{d_{low}^{{WM}_{2}}}\;{uc_{lim}}_{d_{low}^{{WM}_{2}}}\right]$	[36.88 39.54]
**Medium power mode control intervals for program mode 2**	
$\Delta^{c}P_{med}^{{WM}_{2}}=\left[{lc_{lim}}_{P_{med}^{{WM}_{2}}}\;{uc_{lim}}_{P_{med}^{{WM}_{2}}}\right]$	[95.86 235.24]
$\Delta^{c}PF_{med}^{{WM}_{2}}=\left[{lc_{lim}}_{PF_{med}^{{WM}_{2}}}\;{uc_{lim}}_{PF_{med}^{{WM}_{2}}}\right]$	[0.6446 0.7525]
$\Delta^{c}d_{med}^{{WM}_{2}}=\left[{lc_{lim}}_{d_{med}^{{WM}_{2}}}\;{uc_{lim}}_{d_{med}^{{WM}_{2}}}\right]$	[71.93 74.57]
**High power mode control intervals for program mode 2**	
$\Delta^{c}P_{high}^{{WM}_{2}}=\left[{lc_{lim}}_{P_{high}^{{WM}_{2}}}\;{uc_{lim}}_{P_{high}^{{WM}_{2}}}\right]$	[1661.80 2029.00]
$\Delta^{c}PF_{high}^{{WM}_{2}}=\left[{lc_{lim}}_{PF_{high}^{{WM}_{2}}}\;{uc_{lim}}_{PF_{high}^{{WM}_{2}}}\right]$	[0.9701 0.9999]
$\Delta^{c}d_{high}^{{WM}_{2}}=\left[{lc_{lim}}_{d_{high}^{{WM}_{2}}}\;{uc_{lim}}_{d_{high}^{{WM}_{2}}}\right]$	[30.85 33.73]

**Table 3 sensors-22-06639-t003:** Performance metrics of WM-ANN.

Class	Accuracy	Precision	Recall	F1−Score
Anomaly free	96.67%	0.88	1.00	93.75%
Anomaly 1	100.00%	1.00	1.00	100.00%
Anomaly 2	100.00%	1.00	1.00	100.00%
Anomaly 3	99.58%	1.00	0.97	98.59%
Anomaly 4	97.08%	1.00	0.81	89.50%
Anomaly 5	100.00%	1.00	1.00	100.00%
**Micro-avg.**	98.89%	0.9667	0.9667	96.67%
**Macro-avg.**	98.89%	0.9804	0.9630	96.93%
**Weighted-avg.**	98.89%	0.9706	0.9667	96.86%

**Table 4 sensors-22-06639-t004:** Performance metrics of all appliances.

Appliance	Accuracy	Accuracy	F1−Score	F1−Score	F1−Score
	CC-AD	ANN-ADC	CC-AD	ANN-ADC	Hybrid
Washing machine	95.00%	98.89%	94.79%	96.93%	95.53%
Dishwasher	97.12%	98.97%	95.65%	97.10%	96.04%
Refrigerator	96.25%	99.09%	96.78%	98.14%	97.29%
Air conditioner	94.56%	96.34%	93.45%	95.65%	94.22%
Kettle	97.98%	98.14%	97.30%	97.45%	97.42%
Lamp	99.03%	99.45%	98.79%	99.15%	98.92%
**Overall**	97.14%	98.34%	96.21%	97.40%	96.74%

## Data Availability

The data that support the findings of this study are available from the corresponding author, Hanife Apaydin-Özkan, upon reasonable request.

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
