# Peer review of "Appliance-Level Anomaly Detection by Using Control Charts and Artificial Neural Networks with Power Profiles"

_sensors, 2022, doi:10.3390/s22176639_

Round 1

Reviewer 1 Report

Comments to the Author

This paper proposes a novel appliance monitoring and anomaly detection system, which detects anomalies in home appliances before they happen. It is an interesting topic and the paper studies the concept clearly. However, there are several points need to be addressed to improve the quality of the manuscript.

Suggestions to improve the quality of the paper are provided below:

1)     The authors should have a dedicated literature review section to highlight the past works on this topic instead of merging it with the introduction. Within the literature review section, the author should also:

a.     Include a brief literature review of past studies that dealt with appliance-level energy consumption data and the feature generation steps they took to extract the appliances’ power signature for different application areas. Please refer to the studies provided below as a reference.

Plug Load Identification from Low and High Frequency Measurements

Tekler, Z. D., Low, R., Zhou, Y., Yuen, C., Blessing, L., & Spanos, C. (2020). Near-real-time plug load identification using low-frequency power data in office spaces: Experiments and applications. Applied Energy275, 115391.

Gao, J., Kara, E. C., Giri, S., & Bergés, M. (2015, December). A feasibility study of automated plug-load identification from high-frequency measurements. In 2015 IEEE global conference on signal and information processing (GlobalSIP) (pp. 220-224). IEEE.

Load Monitoring for Energy Management

Abubakar, I., Khalid, S. N., Mustafa, M. W., Shareef, H., & Mustapha, M. (2017). Application of load monitoring in appliances’ energy management–A review. Renewable and Sustainable Energy Reviews67, 235-245.

Occupant Behaviour and Plug Load Interactions

Tekler, Z. D., Low, R., & Blessing, L. (2019, September). Using smart technologies to identify occupancy and plug-in appliance interaction patterns in an office environment. In IOP Conference Series: Materials Science and Engineering (Vol. 609, No. 6, p. 062010). IOP Publishing.

b.     Include a literature review of past studies/applications that used the Control Chart method. It would be good to highlight the method’s application in other applications to allow the readers to have a better understanding of their similarities with this work.

2)     In the case of anomaly detection studies, I believe that it is a common practice to select an evaluation metric that places a heavier weight on the model’s performance in abnormal cases over the normal cases due to their unequal distribution. One example includes macro-average f1 score. Please refer to the following literature on those evaluation metrics to include in this work.

Haixiang, G., Yijing, L., Shang, J., Mingyun, G., Yuanyue, H., & Bing, G. (2017). Learning from class-imbalanced data: Review of methods and applications. Expert systems with applications73, 220-239.

Low, R., Cheah, L., & You, L. (2020). Commercial vehicle activity prediction with imbalanced class distribution using a hybrid sampling and gradient boosting approach. IEEE Transactions on Intelligent Transportation Systems22(3), 1401-1410.

Fernández, A., García, S., Galar, M., Prati, R. C., Krawczyk, B., & Herrera, F. (2018). Learning from imbalanced data sets (Vol. 10, pp. 978-3). Berlin: Springer.

3)     In Table 4, it would be nice to perform a comparison between a pure CC-AD-based approach, a pure ANN-AD-based approach, and an AM-ADS approach to see the value of the proposed hybrid-based approach.

4)     The authors should spend some time to discuss about the practicality of the experimental setup and the system’s deployment in real-world conditions. This includes:

a.     Are the findings in Table 2 generalisable to other washing machines, dishwashers, kettle, etc from other brands?

b.     Can the proposed system be applied to other appliance types? Why and why not?

5)     In the conclusion section, the authors made a little effort to propose future directions that could be considered as an improvement on the existing method proposed. I suggest that the authors spend some time thinking deeply about the further improvements of the existing work and highlight some future directions that researchers can benefit from.

Reviewer 2 Report

I think the introduction to the Artificial Neural Network technique is too detailed (section 3.2). It would be enough to point out a good book for readers unfamiliar with this subject. Equations 4, 5, and 6 are not referenced in the text so they can be omitted. Please add descriptions of all used symbols if you decide to leave them. There is a typo in equation 6.

The experiments executed to perform supervised learning are sufficient to collect data for a simple decision algorithm that compares a small number of variables. So, what is the advantage of applying such an advanced and complex technique as ANN?

The article is well written. However, I suggest proofreading. Here are some of the imperfections I noticed:

There are missed white spaces, e.g. in lines 294 and 464.

There are misplaced '%' characters, e.g. in line 483. 

Line 486: "exogenous factors" -> "by exogenous factors".

Line 505: "ANN-ADC method provides much more diverse and comprehensive anomaly classification compared to the similar works in the literature as it can detect not only user usage anomalies but also component anomalies." – Some example evidence should be given.

Line 515: "every appliance can be included in AM-ADS simply with only a smart plug connection without making any mechanical changes; such as, mounting sensors and connections." – However, every smart plug is an IoT device containing sensors and connected to the AM-ADS computer-based system.

Round 2

Reviewer 1 Report

Thank you for addressing my concerns. The current version of the manuscript is now much clearer and structured. 

Author Response

Thank you for the time and effort you dedicated to providing feedback on our manuscript.